# Alpha 1,3 N-Acetylgalactosaminyl Transferase (GTA) Impairs Invasion Potential of Trophoblast Cells in Preeclampsia

**DOI:** 10.3390/ijms25137287

**Published:** 2024-07-02

**Authors:** Yaqi Li, Hongpan Wu, Xiaosong Pei, Shuai Liu, Qiu Yan

**Affiliations:** Liaoning Provincial Core Lab of Glycobiology and Glycoengineering, Department of Biochemistry and Molecular Biology, Dalian Medical University, Dalian 116044, China; liyq01@dmu.edu.cn (Y.L.); wuhp@dmu.edu.cn (H.W.); peixiaosong1996@dmu.edu.cn (X.P.)

**Keywords:** GalNAc α1,3 Gal, α1,3 N-acetylgalactosaminyl transferase, trophoblast cells, DNA methylation, preeclampsia

## Abstract

Preeclampsia (PE) is a pregnancy-specific disorder associated with shallow invasion of the trophoblast cells and insufficient remodeling of the uterine spiral artery. Protein glycosylation plays an important role in trophoblast cell invasion. However, the glycobiological mechanism of PE has not been fully elucidated. In the current study, employing the Lectin array, we found that soybean agglutinin (SBA), which recognizes the terminal N-acetylgalactosamine α1,3-galactose (GalNAc α1,3 Gal) glycotype, was significantly increased in placental trophoblast cells from PE patients compared with third-trimester pregnant controls. Upregulating the expression of the key enzyme α1,3 N-acetylgalactosaminyl transferase (GTA) promoted the biosynthesis of terminal GalNAc α1,3 Gal and inhibited the migration/invasion of HTR8/SVneo trophoblast cells. Moreover, the methylation status of GTA promoter in placental tissues from PE patients was lower than that in the third trimester by methylation-specific PCR (MSP) and bisulfite sequencing PCR (BSP) analysis. Elevated GTA expression in combination with the DNA methylation inhibitor 5-azacytidine (5-AzaC) treatment increased the glycotype biosynthesis and impaired the invasion potential of trophoblast cells, leading to preeclampsia. This study suggests that elevated terminal GalNAc α1,3 Gal biosynthesis and GTA expression may be applied as the new markers for evaluating placental function and the auxiliary diagnosis of preeclampsia.

## 1. Introduction

Preeclampsia is a systemic syndrome characterized by new-onset hypertension and proteinuria with a pregnancy period ≥20 weeks [1,2]. This pregnancy-related disorder often causes adverse outcomes, such as intrauterine growth restriction (IUGR) and fetal death. The dysfunction of extravillous trophoblast cells (EVTs) is considered one of the main causes of PE [3]. EVTs invade the endometrium and maternal spiral artery to facilitate embryo implantation and vascular remodeling. The establishment of maternal–fetal connection guarantees adequate nutrition and oxygen supply for the fetus [4,5]. The impaired migration and invasion of EVTs are related to shallow implantation and abnormal pregnancy. The evaluation of EVT function is meaningful for the early diagnosis and prevention of preeclampsia.

Glycosylation is an essential posttranslational modification of protein molecules that are involved in embryo maturation, uterine receptivity, placental development, etc. The specific glycotypes, known as the “sugar code”, carry important biological information during reproduction [6,7]. Studies revealed that the terminal oligosaccharide structure of glycans (sugar chains), such as sLeX and LeY, facilitates the adhesion behaviors between endometrium and embryo at the maternal–fetal interface [8,9]. The reproductive function of the terminal GalNAc α1,3 Gal structure has not been elucidated.

The biosynthesis of glycans is catalyzed by specific glycotransferases. Alpha 1,3 N-acetylgalactosaminyl transferase (GTA) is a key enzyme controlling terminal GalNAc α1,3 Gal biosynthesis. It transfers UDP-GalNAc to the galactose of the acceptor H antigen to produce A antigen [GalNAc α1,3(Fuc α1,2) Gal β1, 3/4 GlcNAc β1, 3 R] [10]. The abnormal expression or activity of glycotransferases disturbs the corresponding glycan biosynthesis, resulting in pathogenesis. Beta 1,4-galactosyltransferase III (B4GALT3) increased the N-glycan on integrin β1 and suppressed extravillous trophoblast invasion [11]. The upregulation of fucosyltransferase 4 (FUT4) and LeY contributes to trophoblast cell invasion ability [12]. The function of terminal GalNAc α1,3 Gal in the immune modulation of macrophages during infection has been reported [13]. However, the expression traits and function of GTA, as well as its regulatory mechanism in terminal GalNAc α1,3 Gal biosynthesis during reproduction processes, remain unknown.

Growing evidence reveals that improper DNA methylation is correlated with aberrant gene expression and causes reproductive disorders, including IUGR, PE, and recurrent miscarriage [14,15]. Decreased methylation of guanine nucleotide-binding protein subunit alpha-12 (GNA12) in placenta and peripheral blood DNA samples was related to PE development in pregnant women [16]. The hypermethylated Wnt2 gene was found in PE patients compared with that in the healthy pregnant control. DNA methyltransferases (DNMT1, DNMT3A, and DNMT3B) play important roles in regulating DNA methylation status and influence the expression of specific genes. DNMT1 promoted the proliferation, migration, and invasion of extravillous trophoblasts by inhibiting the methylation level of the apelin receptor (APLNR) [17]. DNMT1 enhanced the hypermethylation of the IGF-1 promoter, which was associated with PE [18]. However, whether the methylation of GTA is regulated by DNMTs in PE needs to be explored.

In this study, we found that terminal GalNAc α1,3 Gal biosynthesis and GTA expression were significantly increased in preeclamptic placental tissues. GTA inhibited migration, invasion, proliferation, and vascularization capacity by increasing glycobiosynthesis. The DNA hypomethylation of the GTA promoter upregulated GTA expression in trophoblast cells. These findings suggest that GalNAc α1,3 Gal and GTA may be used as new markers for evaluating placental function and the diagnosis of preeclampsia.

## 2. Results

### 2.1. Increased Terminal GalNAc α1,3 Gal in Preeclamptic Placental Tissues

Using the Lectin array, we found that the level of GalNAc α1,3 Gal, which was recognized and bound by SBA agglutinin in PE patients, was significantly higher than that in the third trimester according to the heatmap (Figure 1A). Normalized fluorescent intensity data showed that SBA agglutinin was significantly increased in PE (Figure 1B). An immunohistochemistry and Lectin blot were performed to evaluate changes in GalNAc α1,3 Gal changes in human first-trimester villi and in placental tissues from the third trimester and PE groups. The immunohistochemistry staining showed that GalNAc α1,3 Gal was mainly expressed in trophoblast cells and vascular endothelial cells, with the highest level occurring in the placental tissues of PE (Figure 1C). A Lectin blot analysis revealed higher GalNAc α1,3 Gal levels in the third-trimester placenta than in first-trimester villi, and the highest level was observed in preeclamptic placental tissue (Figure 1D). GTA is the key enzyme that catalyzes the addition of UDP-GalNAc to the galactose of the acceptor H antigen to produce A antigen [GalNAc α1,3(Fuc α1,2) Gal β1,3/4 GlcNAc β1,3 R] (Figure 1E). We next detected GTA expression in preeclamptic placental tissues.

### 2.2. Expression of GTA Is Upregulated in Preeclamptic Placental Tissues

The transcriptomic data were queried to analyze GTA expression. Using the Gene Expression Omnibus (GEO, GSE102897, n = 3 pairs), we compared gene expression traits between placental tissues from pregnant patients in the third trimester and PE patients. A heatmap and volcano plot were generated, which showed that the expression of genes (DEGs) in PE tissues, especially GTA mRNA (logFC = 1.054, adj.*p* value = 0.006), was significantly higher in the PE group than in the third trimester (Figure 2A). The mRNA level of GTA was higher in the PE group than in the first trimester and third trimester according to q-PCR detection (Figure 2B). Western blot and immunohistochemistry staining showed similar alternations in GTA in the first trimester, third trimester, and preeclamptic placenta (Figure 2C,D). Similar to glycotype, immunohistochemistry staining also showed that GTA was mainly expressed in trophoblast cells and vascular endothelial cells. The results demonstrated that GTA was abnormally elevated in preeclamptic placental tissues, which was closely related to the pathogenesis of preeclampsia.

### 2.3. GTA Promotes the Biosynthesis of Terminal GalNAc α1,3 Gal in Trophoblast Cells

To explore the regulatory role of GTA in the biosynthesis of the corresponding terminal GalNAc α1,3 Gal, GTA siRNA, and GTA cDNA were transfected into HTR8/SVneo cells, respectively. As shown in Figure 3A–C, silencing GTA by the specific siRNA inhibited GTA expression at the mRNA and protein levels, whereas GTA cDNA transfection increased GTA expression by q-PCR and Western blot detection. Lectin blot and immunofluorescence staining showed that silencing GTA inhibited, while GTA cDNA promoted, terminal GalNAc α1,3 Gal biosynthesis (Figure 3D,E).

### 2.4. GTA Affects EMT, Proliferation, and Vascular Remodeling of Trophoblast Cells

We further investigated the effect of GTA on the migration and invasion capacity of trophoblast cells. Early pregnant villus explants were cultured and transfected with GTA siRNA. The distance of exogenous migration of EVTs was photographed and analyzed. The data revealed that the distance of exogenous migration of EVTs transfected with GTA siRNA was significantly increased compared with that of the scramble group, yet GTA cDNA inhibited EVT migration at 24 h (Figure 4A). Transwell assay and Western blot analysis showed that GTA siRNA transfection promoted the migration and invasion of trophoblast cells, the expression of E-cadherin was decreased significantly, and the expression of N-cadherin, vimentin, and snail was increased compared with that in the scramble group. Meanwhile, GTA cDNA transfection had the opposite effect (Figure 4B,C)**.** Vascular formation is one of the main patterns in placental development during pregnancy. The role of GTA and GalNAc α1,3 Gal in trophoblast cell vascularization was investigated. As shown in Figure 4D, employing a tube formation assay, downregulating GTA expression markedly promoted the tube formation of HTR8/SVneo cells, whereas GTA cDNA partially reversed this effect. Moreover, silencing GTA increased the expression of vascular endothelial cell markers, including VE-cadherin, CD34, and CD31 (Figure 4E). Since trophoblast cell proliferation is essential for placental development, we analyzed the effects of GTA on the proliferation capability. EdU incorporation and cell cycle protein detection showed that the proliferation ability of HTR8/SVneo cells was increased after transfection with GTA siRNA, while GTA cDNA impaired proliferation (Figure 4F,G). Collectively, these findings suggested that GTA was necessary for trophoblast cells to undergo EMT, vascular remodeling, and proliferation.

### 2.5. DNA Methylation Status of GTA Promoter in Third Trimester and Preeclamptic Placental Tissues

DNA methylation is an essential epigenetic mechanism regulating the specific gene expression. Here, we investigated the methylation status of GTA in the third trimester and PE placental tissues. The location of the CpG island in the GTA promoter was determined with Methylation Primer design software (http://www.urogene.org/methprimer/; accessed on 10 November 2023.) (Figure 5F). The mRNA levels of DNA methyltransferases (DNMT1, DNMT3A, and DNMT3B) were analyzed. The results showed that the relative mRNA levels of DNMT1 and DNMT3A were decreased in the PE group, but the DNMT3B level was not significantly changed (Figure 5A–C). Notably, there were negative correlations between GTA and DNMT1 (R = −0.53), GTA and DNMT3A (R = −0.29) in the third trimester, and PE tissues (Figure 5D,E). GTA encoded by the *ABO* allele has multiple promoters and transcription start sites: a proximal promoter between −117 bp and +31 bp, an alternative promoter between −667 bp and −336 bp, and a remote promoter region between −832 bp and −667 bp (Figure 5F). Methylation-specific PCR (MSP) was used to semiqualitatively analyze the proximal promoter, alternative promoter, and remote promoter regions. The results showed that in alternative promoter region 1, between −466 bp and −336 bp, the DNA methylation status was decreased in the PE group compared with in third-trimester placental tissues, and there were no significant changes in the other remaining regions in GTA promoter (Figure 5G). The methylation status of each CpG site was determined by bisulfite sequencing PCR (BSP) sequencing. The methylation level of 23 CpG sites in CpG islands between −231 bp and −466 bp was analyzed by bisulfite sequencing PCR in the third-trimester and preeclamptic placental tissues. The results showed that the average methylation rates in the PE group were decreased compared with those in the third-trimester placenta (Figure 5H,I).

### 2.6. Upregulated GTA Expression by Hypomethylation Inhibits Migration/Invasion and Vascularization of Trophoblast Cells

To further clarify the relationship between the DNA methylation status of GTA promoter and GTA expression, HTR8/SVneo cells were treated with different concentrations of DNA methylation inhibitor 5-AzaC, and GTA expression and GalNAc α1,3 Gal biosynthesis were detected. Western blot results showed that GTA expression was significantly higher in the 0.5 μM 5-AzaC group (Figure 6A). Immunofluorescence staining also showed similar changes (Figure 6B). We further detected whether the alternation of GTA induced by 5-AzaC affected GalNAc α 1,3 Gal biosynthesis. Glycotype biosynthesis was increased after 5-AzaC treatment, as shown by immunofluorescence staining and Lectin blot analysis (Figure 6C,D). In addition, the influence of 5-AzaC on migration/invasion and tube formation capacity of trophoblast cells were evaluated. The results showed that 5-AzaC significantly inhibited the EMT behavior of trophoblast cells, while treatment with GTA siRNA and 5-AzaC weakened the inhibitory effect of 5-AzaC (Figure 6E,F). These results suggested that DNA hypomethylation increased GTA expression and GalNAc α1,3 Gal biosynthesis, which inhibited the invasion and vascularization of trophoblast cells.

## 3. Discussion

Accumulating evidence has revealed the roles of glycobiology in reproduction. Aberrant glycosylation (glycotypes and glycoenzymes) may cause dysfunction of trophoblast cells and the placenta, thus leading to placental-derived diseases. In the present study, we screened and identified a significantly increased terminal GalNAc α1,3 Gal glycotype in the placental tissues of PE compared with that in the third-trimester control via Lectin glycomics. Correspondingly, GTA, which is the key enzyme for GalNAc α1,3 Gal biosynthesis, was highly expressed. Upregulated GTA expression inhibited the migration, invasion, proliferation, and vascularization capacity of trophoblast cells. In addition, the DNA hypomethylation of GTA promoter upregulated GTA expression and GalNAc α1,3 Gal biosynthesis in trophoblast cells. These findings suggest that the stage-specific GalNAc α1,3 Gal biosynthesis and GTA expression are closely related to the pathogenesis of preeclampsia. The terminal N-acetylgalactosamine α1,3-galactose may serve as a potential glycobiomarker for evaluating placenta function and the auxiliary diagnosis of preeclampsia.

Different glycosylation modifications catalyzed by the specific glycosyltransferases affect the functions of trophoblast cells. Yamamoto E, Ino K, Miyoshi E, et al. reported that N-acetylglucosaminyltransferase V (GnT-V) promoted the migration and invasion of trophoblast cells [19]. Abnormally downregulated alpha1,6 fucosyltransferase VIII (FUT8) inhibited the proliferation, migration, and invasion of trophoblast cells [20]. The overexpression of GalNAc transferase 2 (GALNT2) suppressed trophoblast cell invasion [21]. In addition, the knockdown of alpha1,3 fucosyltransferase IV (FUT4) reduced the invasion of trophoblast cells by altering MMP activity in trophoblast cells [22]. At present, PE is considered to be caused by the shallow invasion of trophoblast cells and insufficient remodeling of the uterine spiral artery. Soluble fms-like tyrosine kinase inhibitor-1 (sFlt-1) and placental growth factor (PIGF) are related to antiangiogenic factors and disrupt vascular endothelial cell function in PE. The studies also revealed that the invasion of trophoblast cells into the uterus is similar to that of cancer cell invasion into the metastatic tissues to a certain extent. It was found that the increased GTA expression inhibited the invasion and metastasis ability of breast cancer cells [23]. Herein, we found that upregulating GTA expression increased GalNAc α1,3 Gal biosynthesis and inhibited the invasion and vascularization of trophoblast cells, thus contributing to the occurrence of preeclampsia.

Epigenetic analysis provides a new perspective for the diagnosis and treatment of preeclampsia based on DNA methyltransferases and methylation analysis [24]. Compared with normal control, DNMT1 was significantly downregulated in the PE group [17]. Zheng et al. performed whole-genome bisulfite sequencing of placenta DNA and reported the significant downregulation of DNMT1 and DNMT3A and upregulation of methylcytosine dioxygenase TET2 in partially methylated domains of PE tissues [25]. Tang et al. confirmed that DNMT1 mediated the promoter hypermethylation of the HLA class I histocompatibility antigen, alpha chain G (HLA-G) in PE [26]. In this study, the results showed that DNMT1 and DNMT3A were associated with PE in human placental tissues. Aberrant DNA methylation is related to the abnormal expression of reproductive genes in PE. Hypermethylation of Wnt-2 and metalloproteinase inhibitor 3 (TIMP3) were found in PE tissues [27,28]. The hypomethylation of the ADAMTS7 promoter inhibited the migration and invasion of HTR8/SVneo and JEG-3 cells in PE tissues [29].

To further elucidate the regulatory mechanisms of elevated GTA expression in placental tissues of PE, the DNA methylation status of the GTA promoter region was also analyzed. The results revealed that the DNA methylation levels of GTA promoter (alternative promoter 1) were lower in the placental tissues of PE than that of the third trimester by methylation-specific PCR (MSP) and bisulfite sequencing PCR (BSP). Kominato et al. found that the GTA has multiple promoters and transcription start sites: a proximal promoter between −117 bp and +1 bp, an alternative promoter between −667 bp and −336 bp, and a remote promoter region between −832 bp and −667 bp [30,31]. They are involved in regulating the diversity and flexibility of gene expression in both epithelial and erythroid lineages [32,33]. Here, we demonstrated that the methylation of GTA alternative promoter 1, which is between −466 bp and −231 bp, was decreased in preeclamptic placental tissues through methylation-specific PCR and bisulfite sequencing PCR. Furthermore, demethylation by DNA methylation inhibitor 5-AzaC increased GTA expression and GalNAc α1,3 Gal biosynthesis, which inhibited the invasion and vascularization of trophoblast cells.

In summary, elevated terminal GalNAc α1,3 Gal glycotype biosynthesis and key enzyme GTA expression were found in the placental tissues of preeclamptic patients compared with the third-trimester pregnant control. Upregulating the expression of GTA promoted terminal GalNAc α1,3 Gal biosynthesis and inhibited the invasion and vascularization potential of trophoblast cells. Moreover, the methylation status of GTA promoter in placental tissues of PE was decreased, and demethylation enhanced GTA expression and GalNAc α1,3 Gal biosynthesis. The study suggests that elevated terminal GalNAc α1,3 Gal biosynthesis and GTA expression may be applied as the new glycobiological markers for the evaluation of placental function and the auxiliary diagnosis of preeclampsia.

## 4. Materials and Methods

### 4.1. Tissue Collection

Human villus tissues (8 ± 2 weeks of pregnancy) were collected from 10 pregnant women (1st trimester) who underwent voluntary abortion, and human placental tissues from 20 pregnant women (10 cases of PE patients and 10 cases of controls) who received perinatal care in the Department of Obstetrics and Gynecology of the First Affiliated Hospital of Dalian Medical University. The clinical traits of enrolled PE placentas tissues (Appendix A) were in accordance with the 25th edition of Williams Obstetrics and the American College of Obstetricians and Gynecologists guidelines. Considering the shorter gestation period of PE than normal full-term pregnancy, control placentas were from unexplained preterm labor before 37 weeks as matched 3rd trimester. The control group excluded PE, chorioamnionitis, diabetes, and other obstetrics complications. The collected placental tissues from centro-parenchymal of placental chorionic villous were divided into two parts. One part was cut (approximately 1 cm^2^), fixed in 4% formalin for 48 h, and then embedded in paraffin to prepare paraffin tissue slide (4 µm thickness) for subsequent immunohistochemistry. Another part (the remaining tissues) was subjected to RNA and protein extraction. The acquisition of clinical samples was approved by the Ethics Committee of Dalian Medical University, and participants gave written informed consent.

### 4.2. Experimental Reagents and Antibodies

The following chemicals were used: Methylation inhibitor 5-AzaC (HY-10586, MCE, Rahway, NJ, USA); Matrigel (REF:354262, Corning, NY, USA); primary antibodies against CD31 (Cat No. 11265-1-AP), E-cadherin (Cat No. 20874-1-AP), N-cadherin (Cat No. 22018-1-AP), Vimentin (Cat No. 10366-1-AP), Snail (Cat No. 13099-1-AP), and GAPDH (Cat No. 10494-1-AP) (Proteintech, Wuhan, China); primary antibodies against PCNA (AF0261), Cyclin D1(AF0126), and Cyclin E2 (AF2494) (Beyotime, Shanghai, China); GTA antibody (LS-C482291, Life Span, Seattle, WA, USA); CD34 antibody (ab81289, Abcam, Cambridge, UK); VE-cadherin antibody (sc-515467, Santa Cruz, Dallas, TX, USA); and Soybean Agglutinin (SKU:L-1010, VECTOR Lab, Burlingame, CA, USA).

### 4.3. Cell Culture and Transfection

Human chorionic trophoblast cells (HTR8/SVneo) were obtained from American Type Culture Collection (Manassas, VA, USA). HTR8/SVneo cells were cultured in DMEM/F-12 medium containing 10% FBS and placed in a 37 °C cell incubator. The medium was replaced every 2–3 days. When the cell confluence reached approximately 70–80% confluence, the medium was discarded, and the cells were gently washed with PBS and transfected. Solution A: take 250 μL serum-free DMEM/F-12 into 1.5 mL EP tube and add 4 μg GTA cDNA (Genepharma, Plasmid extraction kit Omega, Shanghai, China), 100 pmol GTA siRNA (Genepharma, Shanghai, China), and wait 5 min. Solution B: take 250 μL serum-free DMEM/F-12 into 1.5 mL EP tube, add 2 μL Lipo2000 (#11668030, Invitrogen, Carlsbad, CA, USA), and wait 5 min. Mixing the above A and B solutions, wait another 15 min. Then, add the mixed solution into different treatment groups, and place it in the incubator for 6–8 h.

### 4.4. Real-Time Quantitative PCR

The culture medium was discarded, the cells were washed with PBS 1–2 times, and Trizol Reagent (T9108, Takara, Tokyo, Japan) was added according to the experimental instructions. Isopropanol in RNA extraction was used to precipitate RNA. Total RNA was used to reverse transcription to generate cDNA (AG11601, AGBio, Changsha, China, Evo M-MLV RT Kit). Real-time PCR was performed with the TransStart^®^ Top Green qPCR SuperMix on ABI Prism 7500 Detection system (Applied Biosystem, Foster City, CA, USA). GAPDH was applied as a loading control. Primer of GTA (5′-3′): Forward: GTCTGGGGAGGCACATTCAA; Reverse: GCCCACCATGAAGTGCTTCT. Primer of DNMT1 (5′-3′): Forward: CCGAGCGAGCCAGAGATAGAG; Reverse: GAGATGCCTGCTTGGTGGAATC. Primer of DNMT3A (5′-3′): Forward: AGCACCACGGCACGGAAG; Reverse: ATGGATGGGGACTTGGAGATCAC. Primer of DNMT3B (5′-3′): Forward: GTGTGAGGAGTCCATTGCTGTTG; Reverse: GCTTCCGCCAATCACCAAGTC. Primer of GAPDH (5′-3′): Forward: GGAGTCCACTGGCGTCTTCAC; Reverse: GCTGATGATCTTGAGGCTGTTGTC.

### 4.5. Western Blot Assay

Cells were lysed with RIPA lysis buffer (BD0031, Bioworld, Minneapolis, MI, USA) and incubated on ice for 10 min. The protein concentration was quantified with BCA protein quantitative kit (#23227, Invitrogen, Carlsbad, CA, USA). Equal amounts of protein were separated via 10% SDS-PAGE and electrotransferred onto nitrocellulose membranes. Incubation of the primary antibody at 4 °C overnight: GTA (1:1000); CD31 (1:1000); CD34 (1:1000); VE-cadherin (1:500); E-cadherin (1:1000); E-cadherin (1:1000); Vimentin (1:1000); Snail (1:1000); Cyclin D1 (1:1000); Cyclin E2 (1:1000); PCNA (1:3000); GAPDH (1:4000); SBA lectin (1:3000). The membranes were incubated with secondary antibody at room temperature for 1 h. An enhanced chemiluminescence (ECL) detection system (#1705060, Bio-Rad, Hercules, CA, USA) was used to visualize the immunoreactive bands. The relative protein level was quantified by densitometry and normalized to the GAPDH level using Image J 1.8.0 software (NIH, Bethesda, MD, USA).

### 4.6. Immunofluorescence

Cell climbing films were fixed with 4% paraformaldehyde for 20 min, followed by blocking with goat serum for 1 h. The cells were incubated with GTA antibody (1:200) and agglutinin SBA (1:200) at 4 °C overnight. After washing with PBS, the cells were incubated with FITC or TRITC-conjugated secondary antibody for 1 h at room temperature. Then, the cells were incubated with DAPI (C1002, 1:4000, Beyotime, Shanghai, China) for 20 min at room temperature. Images were observed under a fluorescence microscope (Olympus, BX83, Tokyo, Japan).

### 4.7. Transwell Assay

After adjusting the cell concentration to 2 × 10^5^ with serum-free medium, cells were plated to the Transwell insert of upper chamber without Matrigel for migration assay or with diluted Matrigel in serum-free medium DMEM/F-12 (1:9) for invasion assay. The incubation time was 18 h for migration and 24 h for invasion assay. Cells were fixed with methanol solution for 15 min, followed by staining with crystal violet dye for 15 min. Cells were subsequently photographed and calculated under bright microscope (Olympus, BX83, Tokyo, Japan).

### 4.8. 5-Ethynyl-20-Deoxyuridine (EdU) Incorporation Assay

Cell counting was performed, and the cell concentration was adjusted to 5 × 10^3^ into 96-well plate. EdU labeling medium was prepared at the concentration of 50 μM at 37 °C for 2 h. The cells were fixed with polyformaldehyde for 30 min at room temperature and 2 mg/mL glycine was added to neutralize formaldehyde. Moreover, 0.1% Triton X-100 was used to improve cell membrane permeability. According to the manufacturer’s instructions, 1 × Apollo dye solution was prepared in the cells and shaken at room temperature for 30 min in the dark. After discarding the dye solution, the cells were washed with 0.1% Triton X-100. Hoechst was used to label living cells. The number of proliferative cells was observed and calculated under bright microscope (Olympus, BX83, Tokyo, Japan).

### 4.9. Tube Formation Assay

Matrigel was placed in a 96-well plate in advance, cell counting was performed, and the cell concentration was adjusted to 1 × 10^4^ with serum-free medium, and then the cells were placed in an incubator for 2 h. Tube formation was observed and imaged under the microscope (Olympus, BX83, Tokyo, Japan).

### 4.10. Methylation-Specific PCR and Bisulfite Sequencing PCR

Extraction of the genomic DNA of tissues (L/N7E442H0, Vazyme, Nanjing, China) according to the manufacturer’s instructions, and modified the genomic DNA with bisulfite. Methylation-specific PCR (MSP) was performed for bisulfite-modified DNA according to the instructions (EM101, TIANGEN, Beijing, China). Bisulfite sequencing PCR was performed by Sangon Biotech (Sangon, Shanghai, China). Methylated primer of pair 1 (5′-3′): Forward: TTAAGGTATTAGGGTTACGAGGGGC; Reverse: CGACCATAACTCCGCGTCT. Unmethylated Primer of pair 1 (5′-3′): Forward: GGATAGGGTTTTAAGGTATTAGGGTTATG; Reverse: CCACATCTAATCTCAACCTCCA. Methylated primer of pair 2 (5′-3′): Forward: GTTGGAGGGTTATAGGTTGC; Reverse: AATACCCCTAAAACCCCGTC. Unmethylated Primer of pair 2 (5′-3′): Forward: TTGGAGGGTTATAGGTTGTGG; Reverse: AATACCCCTAAAACCCCATC. Methylated primer of pair 3 (5′-3′): Forward: AGTTTTACGGGTTCGTTTTTTTC; Reverse: ATACCTTAAAACCCTATCCCCG. Unmethylated Primer of pair 3 (5′-3′): Forward: AGTTTTATGGGTTTGTTTTTTTTGT; Reverse: TAATACCTTAAAACCCTATCCCCAC. Methylated primer of pair 4 (5′-3′): Forward: TTTTTTAGTTTTTGTAGTCGTCGTC; Reverse: GAACCCTAAAACCTCTTCGC. Unmethylated Primer of pair 4 (5′-3′): Forward: TTTTAGTTTTTGTAGTTGTTGTGGT; Reverse: CCAAACCCTAAAACCTCTTCAC.

### 4.11. Statistical Analysis

Image Lab (Bio-Rad, Hercules, CA, USA) and GraphPad Prism 6.01^®^ (GraphPad Software Inc. San Diego, CA, USA) were used to process and analyze images and data. All experiments were performed at least three independent times, and the results are shown as the means ± SEM. The analysis of different groups was performed using one-way ANOVA and an unpaired *t*-test. The statistical significance was indicated as follows: * *p* < 0.05, ** *p* < 0.01, and *** *p* < 0.001.

## Figures and Tables

**Figure 1 ijms-25-07287-f001:**
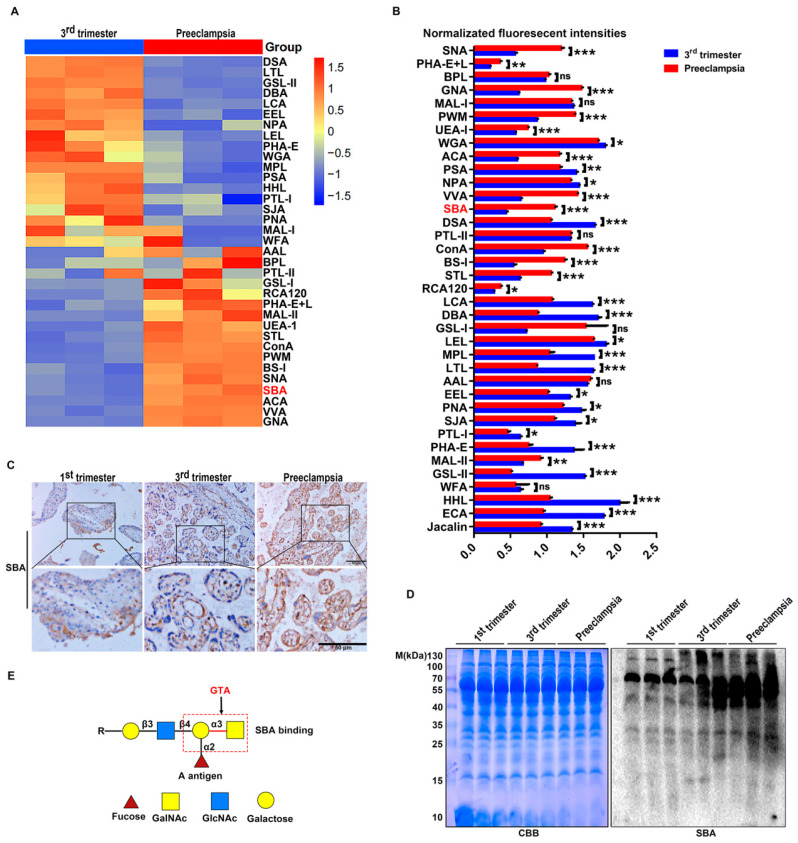
Increased terminal GalNAc α1,3 Gal in preeclamptic placental tissues. (**A**) Heatmap of Lectin clustering analysis in third trimester and PE tissues (n = 3 pairs) by Lectin array. (**B**) Normalized fluorescent intensity analysis of Lectin array. (**C**) Immunohistochemistry detection of GalNAc α1,3 Gal biosynthesis in 1st trimester, 3rd trimester, and PE tissues of human placenta. Black box images are enlargement as lower images. (**D**) Lectin blot of terminal GalNAc α1,3 Gal glycotype recognized by SBA in 1st trimester, 3rd trimester, and PE tissues. Coomassie brilliant blue (CBB): gel staining as equal protein loading. (**E**) Diagram of terminal GalNAc α1,3 Gal biosynthesis catalyzed by GTA. ns means No significance, * *p* < 0.05, ** *p* < 0.01, *** *p* < 0.001.

**Figure 2 ijms-25-07287-f002:**
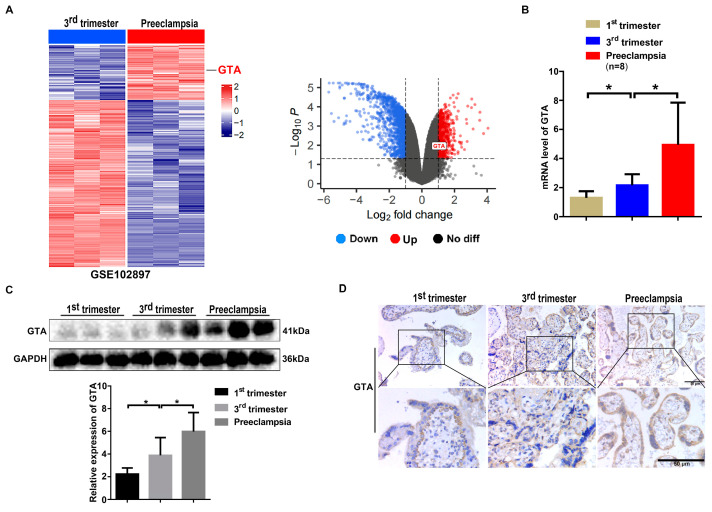
Expression of GTA is upregulated in preeclamptic placental tissues. (**A**) Heatmap and volcano plot of GTA in 3rd trimester tissues and PE tissues from GEO database (GSE102897, n = 3 pairs). (**B**) The mRNA levels of GTA in 1st trimester, 3rd trimester, and PE tissues were detected by q-PCR (n = 8). (**C**) Detection of GTA in 1st trimester, 3rd trimester, and PE tissues by Western blot. (**D**) Immunohistochemistry staining of GTA in 1st trimester, 3rd trimester, and PE tissues of human placenta. Black box images are enlargement as lower images. * *p* < 0.05.

**Figure 3 ijms-25-07287-f003:**
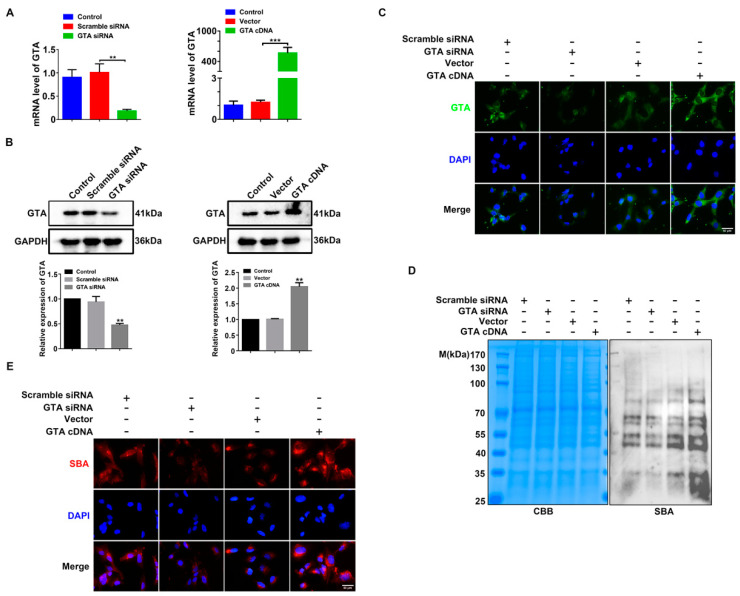
GTA affects the synthesis of GalNAc α1,3 Gal in trophoblast cells. (**A**) Trophoblast HTR8/SVneo cells were transfected with GTA siRNA and GTA cDNA, respectively. The mRNA level of GTA was measured by q-PCR. Western blot (**B**) and immunofluorescence (**C**) were used for GTA intracellular expression and localization. (**D**) Lectin blot of terminal GalNAc α1,3 Gal glycotype recognized by SBA. Coomassie brilliant blue (CBB): gel staining as equal protein loading. (**E**) Immunofluorescence staining of GalNAc α1,3 Gal. Scale bar = 50 µm. DAPI was used for nuclear staining. ** *p* < 0.01, *** *p* < 0.001.

**Figure 4 ijms-25-07287-f004:**
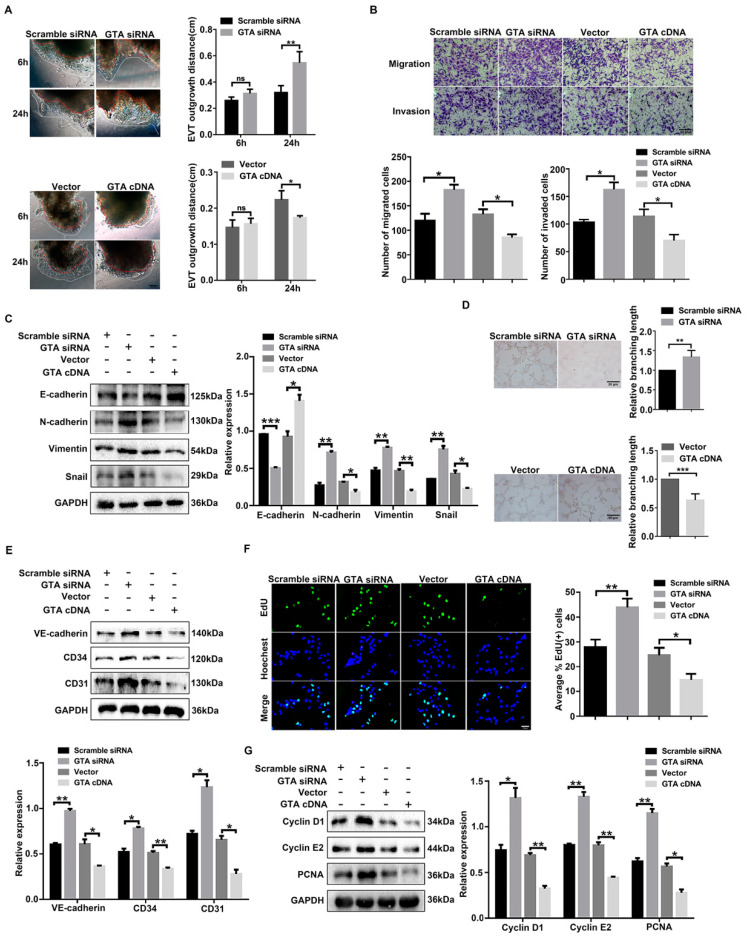
GTA affects EMT, proliferation, and vascular remodeling of trophoblast cells. (**A**) Migration ability of extravillous trophoblast cells (EVTs) was detected by villous explant culture after GTA siRNA and GTA cDNA transfection, respectively. (**B**) Migration and invasion ability of HTR8/SVneo cells was detected by Transwell assay. (**C**) Protein levels of EMT-related proteins (E-cadherin, N-cadherin, Vimentin, and Snail) by Western blot. (**D**) Vascularization potential evaluation by tube formation assay in HTR8/SVneo cells. (**E**) Analysis of vascular markers (VE-cadherin, CD34 and CD31) by Western blot. (**F**,**G**) Proliferation capacity assay by 5-Ethynyl-20-deoxyuridine (EdU) and proliferation-related proteins (Cyclin D1, Cyclin E2, and PCNA) in HTR8/SVneo cells. Scale bar = 50 µm. ns means No significance, * *p* < 0.05, ** *p* < 0.01, *** *p* < 0.001.

**Figure 5 ijms-25-07287-f005:**
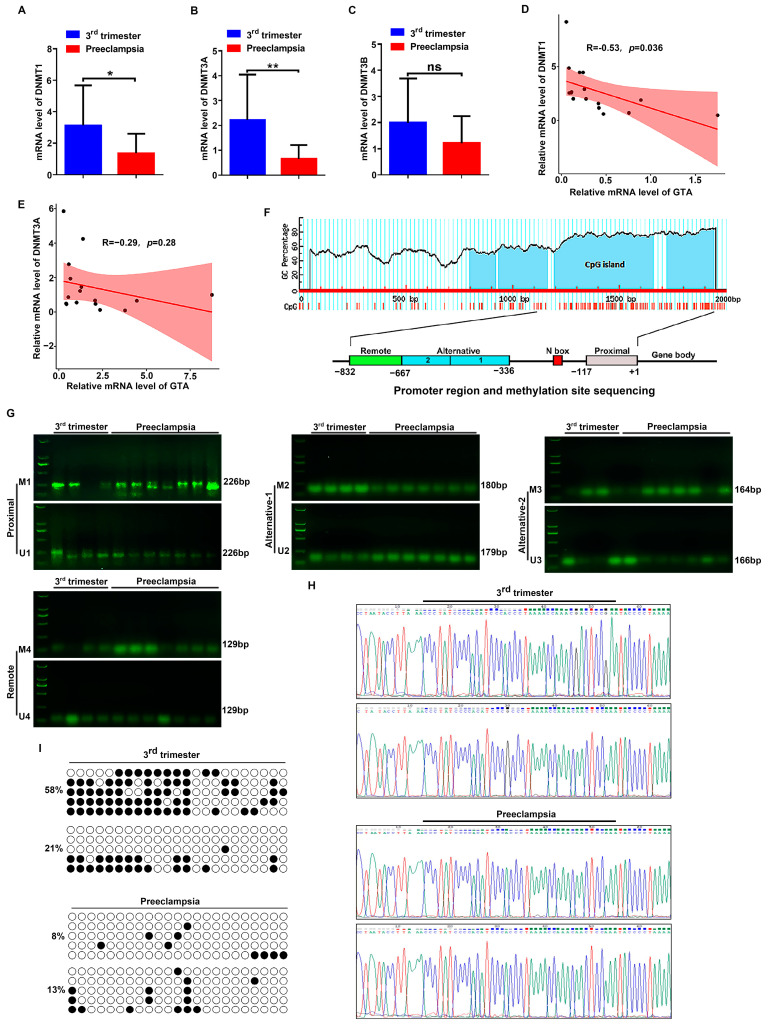
DNA methylation status of GTA promoter in third-trimester and preeclamptic placental tissues. (**A**–**C**) The mRNA levels of DNA methyltransferases (DNMT1, DNMT3A, and DNMT3B) were measured by q-PCR. (**D**,**E**) Correlation analysis between DNMT1 and DNMT3A with GTA expression in 3rd trimester and PE tissues. (**F**) Prediction of CpG islands in GTA promoter region by MethPrimer. (**G**) Methylation analysis of GTA promoter by methylation-specific PCR (MSP) in 3rd trimester and PE tissues. (**H**) Represent display of DNA methylation sequencing. (**I**) Methylation status of each CpG site in five clones by bisulfite sequencing PCR (BSP) in 3rd trimester and preeclamptic placental tissues. White circle: unmethylated CpG site; Black circle: methylated CpG site; * *p* < 0.05, ** *p* < 0.01.

**Figure 6 ijms-25-07287-f006:**
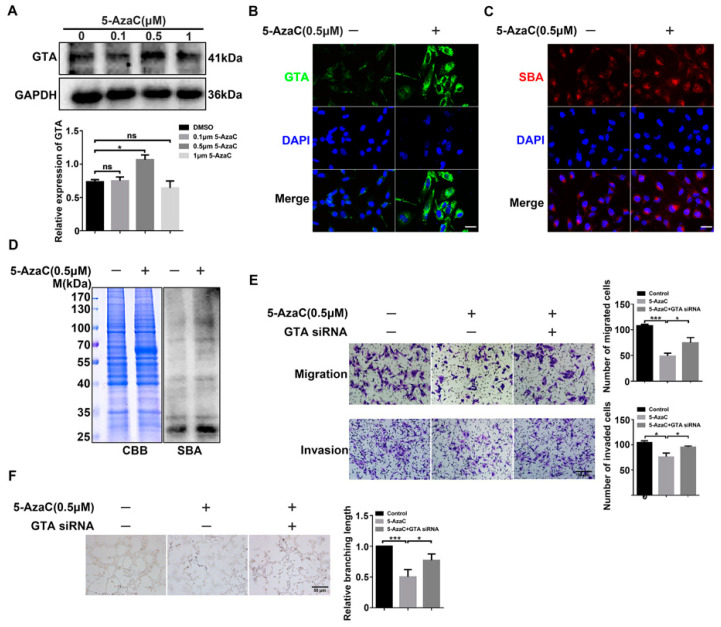
Upregulated GTA expression by hypomethylation inhibits migration/invasion and vascularization of trophoblast cells. (**A**) GTA expression in trophoblast cells after 5-AzaC treatment by Western blot. (**B**) Immunofluorescence staining was used for GTA intracellular expression and localization. DAPI was used for nuclear staining. (**C**) Lectin blot of terminal GalNAc α1,3 Gal glycotype recognized by SBA with 0.5 μM 5-AzaC treatment in HTR8/SVneo cells. Coomassie brilliant blue (CBB): gel staining as equal protein loading. (**D**) Immunofluorescence staining of GalNAc α1,3 Gal. DAPI was used for nuclear staining. (**E**,**F**) Migration, invasion, and vascularization capability of HTR8/SVneo cells after 5-AzaC and GTA siRNA treatment by Transwell assay and tube formation assay. Scale bar = 50 µm. ns means No significance, * *p* < 0.05, *** *p* < 0.001.

## Data Availability

Data supporting the present study are available from the corresponding author upon reasonable request.

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
