# Peer review of "Alpha 1,3 N-Acetylgalactosaminyl Transferase (GTA) Impairs Invasion Potential of Trophoblast Cells in Preeclampsia"

_ijms, 2024, doi:10.3390/ijms25137287_

Round 1

Reviewer 1 Report (Previous Reviewer 2)

Comments and Suggestions for Authors

The Authors presented their original results in the study of the influence of α1,3 N-acetylgalactosaminyl transferase (GTA) on trophoblast cell migration, invasion, and proliferation in preeclampsia (PE). There are grammatical errors, and some sentences are difficult to interpret.

Line 62, "This pregnant disorder," is incorrect. Please change it to "pregnancy disorder" or "pregnancy-related."

Line 95 "including IUGR, PE and so on." Please avoid colloquialism. Either complete the list or rephrase.

Line 96 "For example." Please cancel. The authors stated the same thing in the previous lines.

Line 172 "CD31(Figure 4E)". Missing space.

Lines 274-76 "The results revealed that compared with 3rd trimester, the DNA methylation level of GTA alternative promoter 1 in PE placental tissues was lower semiquantitively methylation specific PCR and bisulfite sequencing". Please clarify this sentence, as it sounds awkward.

Line 277 "Research has found that". Which one? If the Authors were referring to their research, this should be stated. Please rephrase. 

Lines 282-287 and 287-292 "Here, we demonstrated that..." This paragraph has been written twice. Please correct.

I still have major concerns about a few statements:

Line 308 "artificial abortion" Did the Authors mean "voluntary abortion?" or did they refer to endometrial curettage after spontaneous abortion? Please clarify.

Line 309 "Placental tissues were 310 collected from 20 pregnant women who received perinatal care at the Hospital of Dalian 311 Medical University." Were these PE patients? It needs to be clarified by reading the text. Please clarify.

Lines 316 and subsequent "Considering that the 317 gestation period of PE is usually earlier than full term, we collected placentas earlier than 37 weeks as 3rd 318 trimester, from healthy pregnancy with normal biochemical items 319 and physical exams, excluding preeclampsia symptoms" A pregnancy which terminates before 37 weeks of gestation is not healthy by definition. I understand the eagerness of the Author to test a normal control population. I would assume that these pre-term pregnancies may have suffered from different diseases (chorioamnionitis, diabetes, or other obstetrics complications). I would suggest being more transparent about this "control" population.

Lines 319-322: "The placental tissues with the size of 1.5 cm×1.5 cm were selected and embedded in paraffin for immunohistochemistry experiments". Immunohistochemistry is not an "experiment". It is a diagnostic or research tool. Please cancel the "experiment." Was the tissue formalin-fixed, paraffin-embedded? If so, please specify. From which placental area was the used sample chosen? Centro-parenchymal? periferic? 

The rest of the tissues were put into liquid nitrogen, which was used to extract protein and RNA". Which rest of the tissue? How many samples did the Author collect? According to which protocol? Please explain. I guess liquid nitrogen was used not to extract RNA but to freeze tissue to preserve RNA and avoid formalin fixation.

Comments on the Quality of English Language

English is not always readable; there are many grammar errors, and some sentences are difficult to interpret.

Author Response

Response to the reviewer’s comments:

Thank you for reviewing our paper (Title: Alpha 1,3 N-acetylgalactosaminyl transferase (GTA) impairs invasion potential of trophoblast cells in preeclampsia. Manuscript ID: ijms-2768179). We have carefully revised the manuscript following your comments and suggestions, which are very helpful in improving this manuscript. The point-to-point answers are as follows:

Comment 1: Line 62, "This pregnant disorder," is incorrect. Please change it to "pregnancy disorder" or "pregnancy-related."

Answer 1: Thank you for your comment. We have replaced "pregnant disorder" with "pregnancy-related".

Comment 2: Line 95 "including IUGR, PE and so on." Please avoid colloquialism. Either complete the list or rephrase.

Answer 2: Thank you. We have completed the list "including IUGR, PE and recurrent miscarriage."

Comment 3: Line 96 "For example." Please cancel. The authors stated the same thing in the previous lines.

Answer 3: Thank you. We have deleted the "For example."

Comment 4: Line 172 "CD31(Figure 4E)". Missing space.

Answer 4: We have added blank space between "CD31" and "(Figure 4E)".

Comment 5: Lines 274-76 "The results revealed that compared with 3rd trimester, the DNA methylation level of GTA alternative promoter 1 in PE placental tissues was lower semiquantitively methylation specific PCR and bisulfite sequencing." Please clarify this sentence, as it sounds awkward.

Answer 5: Thanks. We have rephrased the sentence as follows: " The results revealed that the DNA methylation levels of GTA promoter (alternative promoter 1) was lower in the placental tissues of PE than that of 3rd trimester by methylation specific PCR (MSP) and bisulfite sequencing PCR (BSP)."

Comment 6: Line 277 "Research has found that". Which one? If the Authors were referring to their research, this should be stated. Please rephrase.

Answer 6: Thanks. We have rephrased the sentence as follows: "Kominato, Y et al. has found that the GTA has multiple promoters and transcription start sites: a proximal promoter between -117 bp and +1 bp, an alternative promoter between -667 bp and -336 bp and a remote promoter region between -832 bp and -667 bp."

Comment 7: Lines 282-287 and 287-292 "Here, we demonstrated that..." This paragraph has been written twice. Please correct.

Answer 7: Sorry for error. We have deleted one.

Comment 8: Line 308 "artificial abortion" Did the Authors mean "voluntary abortion?" or did they refer to endometrial curettage after spontaneous abortion? Please clarify.

Answer 8: Thank you for the comments. "artificial abortion" really means "voluntary abortion". The pregnant women terminated pregnancy voluntarily by endometrial curettage. We replaced "artificial" to "voluntary".

Comment 9: Line 309 "Placental tissues were collected from 20 pregnant women who received perinatal care at the Hospital of Dalian Medical University." Were these PE patients? It needs to be clarified by reading the text. Please clarify.

Answer 9: Sorry for the unclear description. Placental tissues were collected from 20 pregnant women who received perinatal care at the First Affiliated Hospital of Dalian Medical University. Ten of the tissues were from PE patients, and ten from 3rd trimester before 37 weeks excluding PE symptoms and other pregnancy-related diseases.

Comment 10: Lines 316 and subsequent "Considering that the gestation period of PE is usually earlier than full term, we collected placentas earlier than 37 weeks as 3rd trimester, from healthy pregnancy with normal biochemical items and physical exams, excluding preeclampsia symptoms" A pregnancy which terminates before 37 weeks of gestation is not healthy by definition. I understand the eagerness of the Author to test a normal control population. I would assume that these pre-term pregnancies may have suffered from different diseases (chorioamnionitis, diabetes, or other obstetrics complications). I would suggest being more transparent about this "control" population.

Answer 10: We agreed the reviewer’s comments that pregnancy which terminates before 37 weeks of gestation is not healthy by definition. Here, we had carefully analyzed the normal control cases, excluding preeclampsia, chorioamnionitis, diabetes, or other obstetrics complications.

Comment 11: Lines 319-322: "The placental tissues with the size of 1.5 cm×1.5 cm were selected and embedded in paraffin for immunohistochemistry experiments". Immunohistochemistry is not an "experiment". It is a diagnostic or research tool. Please cancel the "experiment." Was the tissue formalin-fixed, paraffin-embedded? If so, please specify. From which placental area was the used sample chosen? Centro-parenchymal? periferic?

Answer 11: Thank you for the comments. We have cancelled the "experiment". The new sentence as follows: The placental tissues were fixed in 5% formalin for 2 days, and then embedded in paraffin according to the standard procedure (4 µm thickness). The placental area was from centro-parenchymal (chorionfrondosum). These useful suggestions were added to revised manuscript.

Comment 12: The rest of the tissues were put into liquid nitrogen, which was used to extract protein and RNA". Which rest of the tissue? How many samples did the Author collect? According to which protocol? Please explain. I guess liquid nitrogen was used not to extract RNA but to freeze tissue to preserve RNA and avoid formalin fixation.

Answer 12: Sorry for unclear statements. The collected placental tissues were divided into three parts. One part was cut (approximately 1 cm2), and fixed in formalin for immunohistochemistry. One part was used to extract RNA and protein. Another part (the rest of tissues) was frozen in liquid nitrogen.

Reviewer 2 Report (Previous Reviewer 3)

Comments and Suggestions for Authors

The authors of this manuscript have tried to elucidate the underlying glycosylated related mechanism during the trophoblastic cell invasion in preeclampsia (PE). They found increased level of terminal GalNAc-a1, 3 Gal in blood group in placental trophoblast cells from preeclampsia patient as compared to healthy individuals with the help of soybean agglutinin lectin. This work demonstrates the importance of GTA mediated overexpression of GalNAc-a 1,3 Gal those results in dysregulated migration an invasion of extravillous trophoblast cells (EVTs). With the figures and some more information added the study seems well presented now. However, I have some concerns and questions about some of the results presented in this manuscript. The manuscript needs to edited for general English before it can be published. There are multiple instances where, the sentences are convoluted and should be rewritten so that it reads easy and makes more sense to the readers.

Line 64: Dysfunctional should be changed to “dysfunction”.

Line 89: “The function of terminal GalNAc α 1,3 Gal on immune modulation of macrophages during infection has been reported.” Authors should provide a reference for this statement here.

Figure 1a. Authors have looked at GalNAc-a1, 3-Gal expression in the 3rd trimester and Preeclampsia using the SBA agglutinin. I was wondering if author also looked at 1st trimester samples too.

Figure 1D. The signal for the lectin blot in 3rd trimester samples does not seem to be even. The authors did not explain the reason behind this discrepancy.

Line 125: “3rd trimester placenta than in 1st villi”. It should be 1st “trimester” villi.

Figure 2C: Western blot with GTA: the 3 lanes in 3rd trimester samples have inconsistent intensity for GTA, just like the figure 1d. Is there a reason for these differences?

Figure 2D. Please clarify if the scale bars are same for all the panels?

Figure 6A. The authors tested the 5-AzaC at different concentration (0.1 to 1 uM) and they start to see no effect at 1 uM concentration. I was wondering if authors tested other concentrations between 0.1 uM to 0.5 uM to see if they find a concentration that shows even better effect on GTA expression?

Figure 6 E/F:  HTR8/SVneo cells when co-treated with GTA siRNA and 5-AzaC did not rescue the EMT behavior of trophoblast cells to the levels of control. Is it possible that other GalNAc transferases may also be playing role in dysfunction of trophoblast cells and placenta and hence leading to preeclampsia?

Author Response

Thank you for reviewing our paper (Title: α1,3 N-acetylgalactosaminyl transferase (GTA) influences migration, invasion and proliferation of trophoblast cells in preeclampsia. Manuscript ID: ijms-2664419). We have carefully revised the manuscript following your comments and suggestions, which are very helpful in improving this manuscript. The point-to-point answers are as follows: 

Comment 1: Line 64: Dysfunctional should be changed to “dysfunction”.

Answer 1: We have changed “dysfunctional” to “dysfunction”.

Comment 2: Line 89: “The function of terminal GalNAc α 1,3 Gal on immune modulation of macrophages during infection has been reported.” Authors should provide a reference for this statement here.

Answer 2: Thanks. The reference was added:

[33] Zhou R, Wang X, Liu H, et al. GalNAc-Specific Soybean Lectin Inhibits HIV Infection of Macrophages through Induction of Antiviral Factors. J Virol. 2018;92(6):e01720-17.

Comment 3: Figure 1a. Authors have looked at GalNAc-a1, 3-Gal expression in the 3rd trimester and Preeclampsia using the SBA agglutinin. I was wondering if author also looked at 1st trimester samples too.

Answer 3: Thank you for your comments. We screened the glycotypes by Lectin array in 3rd trimester and preeclampsia, and aimed to find differential glycotypes related to preeclampsia pathogenesis according to clinical significance. Placental function in 1st trimester usually affects pregnancy outcomes, so we detected the glycotypes GalNAc α1, 3 Gal expression levels in 1st trimester by immunochemistry and Lectin blot (Figure 1C&D).

Comment 4: Figure 1D. The signal for the lectin blot in 3rd trimester samples does not seem to be even. The authors did not explain the reason behind this discrepancy.                                                         Answer 4: The reviewer’s comment is right. Three samples in 3rd trimester were not even. In fact, we detected 10 samples of 3rd trimester, majority of them were high expression, less of them were low levels. Here, we loaded the representative samples in Figure 1D, and the average level of GalNAc α1, 3 Gal was higher in 3rd trimester compared with 1st trimester. Similar to GalNAc α1, 3Gal, the representative samples of GTA expression were loaded in Figure 2D.

Comment 5: Line 125: “3rd trimester placenta than in 1st villi”. It should be 1st “trimester” villi.
Answer 5: Thanks. We have corrected it.

Comment 6: Figure 2C: Western blot with GTA: the 3 lanes in 3rd trimester samples have inconsistent intensity for GTA, just like the figure 1d. Is there a reason for these differences?

Answer 6: We agreed with the reviewer’s comments. As mentioned before, three samples in 3rd trimester was not even. In fact, we also detected 10 samples of 3rd trimester, majority of them were high expression, less of them were low levels. Here, we loaded the representative samples in Figure 2D, and the average expression of GTA was higher in 3rd trimester compared with 1st trimester.

Comment 7: Figure 2D. Please clarify if the scale bars are same for all the panels?

Answer 7: Thank you for your comments. In Figure 2D, representative area of the top images were enlarged. We added the scale bars for all the panels.

Comment 8: Figure 6A. The authors tested the 5-AzaC at different concentration (0.1 to 1 μM) and they start to see no effect at 1 μM concentration. I was wondering if authors tested other concentrations between 0.1 μM to 0.5 uM to see if they find a concentration that shows even better effect on GTA expression?

Answer 8: Thank you for your comments. We detected GTA expression under different concentrations between 0.1 μM to 0.5 μM by Western blot. We found that GTA expression was significantly higher in 0.5 μM 5-AzaC group by double check.

Comment 9: Figure 6 E/F:  HTR8/SVneo cells when co-treated with GTA siRNA and 5-AzaC did not rescue the EMT behavior of trophoblast cells to the levels of control. Is it possible that other GalNAc transferases may also be playing role in dysfunction of trophoblast cells and placenta and hence leading to preeclampsia?

Answer 9: Thank you for your comments. Our results demonstrated that GTA siRNA could promote the EMT behaviors of HTR8/SVneo cells (Figure 4), 5-AzaC inhibited the EMT behaviors of HTR8/SVneo cells, but when co-treated with GTA siRNA and 5-AzaC could restore the EMT behaviors of HTR8/SVneo cells compared with 5-AzaC group (Figure 6E, F). 5-AzaC specifically inhibits DNA methylation, and contributes to reverse epigenetic changes of EMT-related genes, including GTA. The dosage of GTA siRNA varied fellow to the number of cells according to the operating manual in our experiments. However, GTA siRNA could not completely reverse the inhibitory effect by 5-AzaC. There was partial rescue in Transwell assay and tube formation experiment.

We have searched Enzyme databases (www.enzyme-database.org) and literatures. Although there are 11 GalNAc transferases related to eukaryotes, such as glycoprotein-fucosylgalactoside α-N-acetylgalactosaminyltransferase (GTA), polypeptide N-acetylgalactosaminyltransferase  (ppGalNAcTs /GALNTs), protein O-mannose β-1,3-N-acetylgalactosaminyltransferase (B3GALNT2), globotriaosylceramide 3-β-N-acetylgalactosaminyltransferase (β1-3GalNAcT), globoside α-N-acetylgalactosaminyltransferase (α1-3 GalNAcT/ Forssman synthase), (N-acetylneuraminyl)-galactosylglucosylceramide N-acetylgalactosaminyltransferase (GM2 synthase/ B4GALNT1). These GalNAc transferases transfer GalNAc from the donor substrate (UDP-GalNAc) to the acceptor substrates to produce with different glycotypes. ppGalNAcTs could transfer GalNAc to threonine or serine hydroxy groups on the polypeptide core of submaxillary mucin, κ-casein, apofetuin and some other acceptors of high molecular mass. GTA catalyzed terminal GalNAc α 1,3 Galbiosynthesis in blood group A. According to Lectin array detection, SBA recognized GalNAc α1,3 Gal terminal structure, which atalyzed by GTA. Here, we select GTA for further study.

Reviewer’s suggestion is good, we will further explore the function and the mechanism of other GalNAc transferases in preeclampsia in future work. Thanks.

Round 2

Reviewer 1 Report (Previous Reviewer 2)

Comments and Suggestions for Authors

The paper was improved in response to the last revision, although the control cases remain a significant issue.

Major concerns

Lines 306-7; 318-19 I can't understand why the Authors go on without explaining clearly and transparently which are those "healthy" control cases whose pregnancy terminated before 37th weeks of gestation but that wasn't affected by "PE, chorioamnionitis, diabetes, or other obstetrics complications." The control case population characteristics have not been clarified, as previously asked.

For clarity's sake, please write about the controls only once and not in different parts of the same paragraph.

Minors

Line 313 "to≥140" add a space after "to"

Line 315 "of≥300" add a space after "of"

Line 320 "chorion frondosum" is incorrect outside the embryonal period of placentation. Did you include both the fetal and the maternal surfaces? As PE is a pathology of the fetal-maternal interface, that would be of particular importance.

Lines 321-22 "The placental area was from centro-parenchymal (chorion frondosum). One part was cut (approximately 1 cm2), fixed in 5% formalin for 2 days". The standard % of formalin fixation is usually 4%. Any reason why the Authors chose 5%? As the average fixation time in surgical pathology is 24 hours, is there any reason why the Authors cut the fresh placentas and then fixed the samples for 48 hours? Otherwise, the placentas were entirely formalin-fixed and then sampled. Please clarify this point, as it is not clear at all. 

Lines 321-22 "then embedded in paraffin according to the standard procedure (4 µm thickness)". I think this refers to the thickness of the tissue slide (4 µm thickness) and not to the paraffin embedding. Please rephrase and clarify.

Author Response

Response to the reviewer’s comments:

Thank you for reviewing our paper (Title: Alpha 1,3 N-acetylgalactosaminyl transferase (GTA) impairs invasion potential of trophoblast cells in preeclampsia. Manuscript ID: ijms-2768179). We have carefully revised the manuscript following your comments and suggestions, which are very helpful in improving this manuscript. The point-to-point answers are as follows:

Comment 1: Lines 306-7; 318-19 I can't understand why the Authors go on without explaining clearly and transparently which are those "healthy" control cases whose pregnancy terminated before 37th weeks of gestation but that wasn't affected by "PE, chorioamnionitis, diabetes, or other obstetrics complications." The control case population characteristics have not been clarified, as previously asked. For clarity's sake, please write about the controls only once and not in different parts of the same paragraph.

Answer 1: Thanks for your comments. We have seriously consulted the clinicians again, and our statements did was unclear. We have rewritten the paragraph as follows:

Human villus tissues (8±2 weeks of pregnancy) were collected from 10 pregnant women (1st trimester) who underwent voluntary abortion, and human placental tissues from 20 pregnant women (10 cases of PE patients and 10 cases of controls) who received perinatal care in the Department of Obstetrics and Gynecology of the First Affiliated Hospital of Dalian Medical University. The clinical traits of enrolled PE placentas tissues (supplementary) were in accordance with the 25th edition of Williams Obstetrics, and the American College of Obstetricians and Gynecologists guidelines. Considering the shorter gestation period of PE than normal full-term pregnancy, control placentas were from unexplained preterm labor before 37 weeks as matched 3rd trimester. The control group excluded PE, chorioamnionitis, diabetes, and other obstetrics complications. The collected placental tissues from centro-parenchymal of placental chorionic villous, were divided into two parts. One part was cut (approximately 1 cm2), fixed in 4% formalin for 48 h and then embedded in paraffin to prepare paraffin tissue slide (4 µm thickness) for subsequent immunohistochemistry. Another part (the remaining tissues) was subjected to RNA and protein extraction. The acquisition of clinical samples was approved by the Ethics Committee of Dalian Medical University, and participants gave written informed consent.

Comment 2: Line 313 "to≥140" add a space after "to"

Line 315 "of≥300" add a space after "of"

Answer 2: Thank you. We have rewritten this paragraph.

Comment 3: Line 320 "chorion frondosum" is incorrect outside the embryonal period of placentation. Did you include both the fetal and the maternal surfaces? As PE is a pathology of the fetal-maternal interface, that would be of particular importance.

Answer 3: Thank you for your comments. It’s true that PE is a pathology of the fetal-maternal interface. The collected placental tissues were mainly from centro-parenchymal of chorionic villous, with a little from maternal decidua. And "chorion frondosum" was deleted in revised manuscript.

Comment 4: Lines 321-22 "The placental area was from centro-parenchymal (chorion frondosum). One part was cut (approximately 1 cm2), fixed in 5% formalin for 2 days". The standard % of formalin fixation is usually 4%. Any reason why the Authors chose 5%? As the average fixation time in surgical pathology is 24 hours, is there any reason why the Authors cut the fresh placentas and then fixed the samples for 48 hours? Otherwise, the placentas were entirely formalin-fixed and then sampled. Please clarify this point, as it is not clear at all.

Answer 4: The placental tissues fixed in 4% formalin, sorry for the error. As reviewer mentioned, fresh tissues were fixed in 4% formalin for 24 h generally. In this study, the placental tissues were fixed for 48 h to make sure the tissues fixed completely.

Comment 5: Lines 321-22 "then embedded in paraffin according to the standard procedure (4 µm thickness)". I think this refers to the thickness of the tissue slide (4 µm thickness) and not to the paraffin embedding. Please rephrase and clarify.

Answer 5: Thanks for your comments. The collected placental tissues (approximately 1 cm2) were fixed in 4% formalin for 48 h, embedded in paraffin, and then prepared paraffin tissue slide (4 µm thickness) for subsequent immunohistochemistry.

Round 3

Reviewer 1 Report (Previous Reviewer 2)

Comments and Suggestions for Authors

The Authors answered all the addressed issues except for the most important one:

"control placentas were from unexplained preterm labor before 37 weeks as matched 3rd trimester. The control group excluded PE, chorioamnionitis, diabetes, and other obstetrics complications. "

The absence of knowledge about the cause of preterm delivery introduces unknown bias in the control population. How can the Author figure out how to reproduce this experiment if they are unaware of the clinical and pathological characteristics of the control population? Moreover, they stated that the control population excluded PE, chorioamnionitis, diabetes, and other obstetric complications (I imagine, for example, placenta previa...) that account for nearly 100% of causes of preterm deliveries.

As previously and repeatedly asked, the authors must provide an identifiable and reproducible control population. As the Authors excluded chorioamnionitis (which can be confirmed only by histology), they can access the histological report. It is not permissible to define "unknown" as the cause of the preterm deliveries they used as a control population.

Histologically proven chorioamnionitis and preterm deliveries due to obstetric complications (such as placenta previa) may be used as surrogates for "non-PE control cases." That would be much more scientifically correct than using a control population with unknown diagnoses.

This manuscript is a resubmission of an earlier submission. The following is a list of the peer review reports and author responses from that submission.

Round 1

Reviewer 1 Report

Comments and Suggestions for Authors

General comments: 

The work presented by Yaqi Li and al. has a merit since it is devoted to the preclampsia which is one of the major pregnancy-related disorders worldwide. However, the way the study is presented from the introduction onwards lacks of clarity. This reviewer did not find in the submitted manuscript, neither in the abstract nor the body of the manuscript, the sentences describing clearly the pursued objectives in this study, and the tested hypotheses by the authors in the undertaken study. These flaws are unfortunate as it became very difficult to assess the potential (or the intended) contribution of the authors to the research field dealing with preclampsia.

Obviously, the manuscript could benefit of the english editing since there are quite a lot of mistakes. Some sentences are too long, and it is therefore difficult to understand the conveyed message. The abbreviations are not systematically defined at their first use. 

The current reviewer was unable to carefully crosscheck the described data and the figures referenced since in the uploaded figure file, not all figures and appropriate panels, were present in the uploaded file. Therefore, this reviewer was unable to appreciate the accurateness and the validity of all statements contained in this paper. This is mainly the case when the authors refer to the missing figures (from the uploaded file) such as Fig 3A, Fig 4A and B. Consequently, since the current reviewer didn't have access to the entire data, it is difficult for the reviewer to evaluate the contribution of the current study to the preclampsia research field (respectively to pregnancy-related diseases' research field.   

The summary should be rewritten, and the conclusions should answer whether the precise undertaken hypotheses/objectives in this study were fulfilled or not. Optionally, a few words on the perspective/follow-up study can be also provided. 

Based on all aforementioned problems, the current reviewer is able to access the real merit and significance of the submitted manuscript. 

specific comments:

line 94: please replace "in" by "during"

Line 95-96: please clarify what do you mean by "human sheep red blood cells"

line 115: the abbreviation APLNR is not defined at first use. 

Line 117-118, please delete once the word "aberrant"

Line 122: please change from "preeclampsia" to "preeclamptic"

Line 271-275: Very long sentence. Please, split at least into two sentences.

Line 282-283 : the authors wrote "GalNAc a 1,3 gal and GTA may serve as a new marker for the diagnosis of preeclampsia from the perspective of glycobiology". Maybe the authors could consider this sentence and write in upstream sections (abstract and introduction) something like " Our objective was to verify whether GalNAc (....) may be used as a new marker for the diagnosis of preeclampsia". 

Line 285: please, change from "provides" to "provide"

Line 313-316: this sentence is not understandable to this reviewer

Author Response

Comment 1: The work presented by Yaqi Li and al. has a merit since it is devoted to the preeclampsia which is one of the major pregnancy-related disorders worldwide. However, the way the study is presented from the introduction onwards lacks of clarity. This reviewer did not find in the submitted manuscript, neither in the abstract nor the body of the manuscript, the sentences describing clearly the pursued objectives in this study, and the tested hypotheses by the authors in the undertaken study. These flaws are unfortunate as it became very difficult to assess the potential (or the intended) contribution of the authors to the research field dealing with preeclampsia.

Answer 1: Thank you for the comments. We have rewritten the introduction and objectives more clearly. we have uploaded again the manuscript abstract and so on.

In current study, we aimed to explore the glycobiology mechanism of preeclampsia pathogenesis. We first screened the obvious glycans by Lectin array in normal pregnancy (3rd trimester) and preeclampsia. Based on the change of glycans (SBA binding GalNAc α1,3 Gal), we identified α1,3 N-acetylgalactosaminyl transferase (GTA), and confirmed its regulation and function in preeclampsia. The results revealed that terminal GalNAc α1,3 Gal and biosynthesis key enzyme GTA were significantly elevated in placenta of preeclampsia, compared with normal pregnancy. Furthermore, hypomethylation of the GTA promoter elevated the expression of GTA, and increased GalNAc α1,3 Gal biosynthesis, which impaired trophoblast cells migration, invasion and proliferation, leading to preeclampsia.

Comment 2: Obviously, the manuscript could benefit of the English editing since there are quite a lot of mistakes. Some sentences are too long, and it is therefore difficult to understand the conveyed message. The abbreviations are not systematically defined at their first use.

Answer 2: The English of manuscript has been modified and long sentences have been shortened. We systematically defined the abbreviations at their first use.

Comment 3: The current reviewer was unable to carefully crosscheck the described data and the figures referenced since in the uploaded figure file, not all figures and appropriate panels, were present in the uploaded file. Therefore, this reviewer was unable to appreciate the accurateness and the validity of all statements contained in this paper. This is mainly the case when the authors refer to the missing figures (from the uploaded file) such as Fig 3A, Fig 4A and B. Consequently, since the current reviewer didn't have access to the entire data, it is difficult for the reviewer to evaluate the contribution of the current study to the preeclampsia research field (respectively to pregnancy-related diseases' research field.  

Answer 3: We have reuploaded the manuscript and figures again in revised version, and make sure the uploaded were complete.

Comment 4: The summary should be rewritten, and the conclusions should answer whether the precise undertaken hypotheses/objectives in this study were fulfilled or not. Optionally, a few words on the perspective/follow-up study can be also provided.

Answer 4: The reviewer’s suggestion is very helpful. We have rewritten the summary and conclusions. The new summary as follows:

Preeclampsia (PE) is a pregnancy-specific disorder associated with shallow invasion of the trophoblast and insufficient remodeling of the uterine spiral artery. Protein glycosylation plays an important role during trophoblast invasion. However, the glycobiological mechanism in PE has not been fully elucidated. In the current study, employing Lectin array, we found that soybean agglutinin (SBA), which recognizes the terminal GalNAc α1,3 Gal glycotype, was significantly increased in the placental tissues from preeclampsia patients compared with 3rd trimester pregnancy controls. Upregulating the expression of a key enzyme, α1,3 N-acetylgalactosaminyl transferase (GTA), promoted the biosynthesis of terminal GalNAc α1,3 Gal and inhibited the migration and invasion of HTR8/SVneo trophoblast cells. Moreover, the methylation status of the GTA promoter in placental tissues of PE was lower than that in the 3rd trimester by methylation specific PCR (MSP) and bisulfite sequencing PCR (BSP) analysis. The elevated GTA expression with the DNA methylation inhibitor 5-azacytidine (5-AzaC) treatment increased the glycotype biosynthesis, and impaired the invasion potential of trophoblast cells, leading to preeclampsia. The study suggests that elevated terminal GalNAc α1,3 Gal biosynthesis and GTA expression may be applied as the new markers for the evaluation of placental function and auxiliary diagnosis of preeclampsia.

Comment 5: Based on all aforementioned problems, the current reviewer is able to access the real merit and significance of the submitted manuscript.

Answer 5: We have uploaded the modified manuscript according to the reviewer’s suggestion.

Comment 6: Line 94: please replace "in" by "during"

Answer 6: Thank you for your comments. We have replaced "in" with "during". The new sentence as follows: Glycoproteins presenting GalNAc are involved in immune modulation during the fertilization process.

Comment 7: Line 95-96: please clarify what do you mean by "human sheep red blood cells"

Answer 7: Sorry for the mistakes, we have corrected it into "The function of terminal GalNAc α 1,3 Gal on immune modulation of macrophages during infection has been reported."

Comment 8: Line 115: the abbreviation APLNR is not defined at first use.

Answer 8: Thank you. We have added the abbreviation "APLNR, Apelin receptor".

Comment 9: Line 117-118, please delete once the word "aberrant"

Answer 9: Thanks. We have deleted one.

Comment 10: Line 122: please change from "preeclampsia" to "preeclamptic"

Answer 10: Line 122: change from "preeclampsia" to "preeclamptic". The other preeclampsia in manuscript was also replaced.

Comment 11: Line 271-275: Very long sentence. Please, split at least into two sentences.

Answer 11: The long sentence is shortened as: Our results found GTA was highly expressed in placental tissues of PE compared with that in 3rd trimester.

Comment 12: Line 282-283: the authors wrote "GalNAc a 1,3 gal and GTA may serve as a new marker for the diagnosis of preeclampsia from the perspective of glycobiology". Maybe the authors could consider this sentence and write in upstream sections (abstract and introduction) something like " Our objective was to verify whether GalNAc (....) may be used as a new marker for the diagnosis of preeclampsia".

Answer 12: We have added the sentence by reviewer in the abstract and introduction.

Comment 13: Line 285: please, change from "provides" to "provide"

Answer 13: We have replaced "provides" with "provide".

Comment 14: Line 313-316: this sentence is not understandable to this reviewer

Answer 14: The sentence has been rewritten as follows: Because 5-AzaC is a DNA methyltransferase inhibitor, it could inhibit DNA methylation specifically. When we treated trophoblast cells with 5-AzaC at an appropriate concentration (0.5 μM), the results showed that GalNAc α 1,3 Gal biosynthesis and GTA expression were increased in trophoblast cells, which inhibited trophoblast epithelial mesenchymal transformation behaviors.

Reviewer 2 Report

Comments and Suggestions for Authors

The Authors presented their original results in the study of the influence of α1,3 N-acetylgalactosaminyl transferase (GTA) on trophoblast cell migration, invasion, and proliferation in preeclampsia (PE). The study is well-presented, however, I have some significant concerns about tissue collection. 

The Authors stated (line 333 and subsequent, Material and Methods section) "Human placental villus tissue (6-10 weeks of pregnancy) was collected from pregnant women (1st trimester) … Placental tissue samples of the 3rd trimester and preeclampsia in the normal pregnancy (≥28 weeks) were collected from cesarean section pregnant women who were hospitalized for delivery”. There are many differences between a 6 weeks and a 10 weeks pregnancy (putting aside that it is quite surprising thinking about a voluntary abortion at 6 weeks, when a mother is barely aware of being pregnant), as the first trophoblast invasion of the endometrium starts at about 8 weeks and the second one at about 12 weeks. Why did the Authors stop the first trimester at 10 weeks? Regarding the placenta of the 3rd trimester, the Authors seem unaware that a placenta can be difficultly considered “normal” when the delivery is before 37 weeks (pre-term delivery). The Authors should present an accurate and detailed report (such as a table) to describe the effective weeks of gestation of the collected tissue sample (voluntary abortions, pPE’s placentas, and normal ones). Moreover, the unawareness of other maternal conditions affecting those pregnancies must be resolved. A table with maternal and pregnancy characteristics should be presented, as  PE is a spectrum, not a single pathology, varying from very mild to extremely severe and deadly. 

line 275 “Emerging study has suggested that GTA exerts an anti-tumor effect in breast cancer [19]. So we speculated that excessive GalNAc α 1,3 Gal and GTA in the placenta may contribute to the cause of preeclampsia”. Please clarify which is the connection with anti-tumor effect in breast cancer and the cause of PE.

line 333 “Placental tissue samples of the 3rd trimester and preeclampsia in the normal pregnancy (≥28 weeks) were collected from cesarean section pregnant women who were hospitalized for delivery” must be rephrased and clarified

Comments on the Quality of English Language

The Paper is quite weel written and clearly comprehensible.

Author Response

Comment 1: The Authors presented their original results in the study of the influence of α1,3 N-acetylgalactosaminyl transferase (GTA) on trophoblast cell migration, invasion, and proliferation in preeclampsia (PE). The study is well-presented, however, I have some significant concerns about tissue collection.

Answer 1: Thank you for the comments. We will modify the manuscript according to the reviewer’s suggestions.

Comment 2: (1) The Authors stated (line 333 and subsequent, Material and Methods section) "Human placental villus tissue (6-10 weeks of pregnancy) was collected from pregnant women (1st trimester) … Placental tissue samples of the 3rd trimester and preeclampsia in the normal pregnancy (≥28 weeks) were collected from cesarean section pregnant women who were hospitalized for delivery”. There are many differences between a 6 weeks and a 10 weeks pregnancy (putting aside that it is quite surprising thinking about a voluntary abortion at 6 weeks, when a mother is barely aware of being pregnant), as the first trophoblast invasion of the endometrium starts at about 8 weeks and the second one at about 12 weeks. Why did the Authors stop the first trimester at 10 weeks? (2) Regarding the placenta of the 3rd trimester, the Authors seem unaware that a placenta can be difficultly considered “normal” when the delivery is before 37 weeks (pre-term delivery). (3) The Authors should present an accurate and detailed report (such as a table) to describe the effective weeks of gestation of the collected tissue sample (voluntary abortions, PE’s placentas, and normal ones). Moreover, the unawareness of other maternal conditions affecting those pregnancies must be resolved. A table with maternal and pregnancy characteristics should be presented, as PE is a spectrum, not a single pathology, varying from very mild to extremely severe and deadly.

Answer 2: (1) We agree with the reviewer’s opinions that there are many differences between 6 weeks and 10 weeks pregnancy. It’s true that two stages (the first trophoblast invasion at 8 weeks and the second at 12 weeks). We aimed to study the glycobiological mechanism in earlier time, therefore, we selected most of villi tissues were around 8 weeks, with a few from 10 weeks and 6 weeks.

(2) We agree with the reviewer’s comments. Considering that the gestation period of PE is usually earlier than full term, we collected placentas earlier than 37 weeks as 3rd trimester, from healthy pregnancy with normal biochemical items and physical exam, excluding preeclampsia symptoms.

(3) As suggested by review, we added a new table about details of tissue samples as supplementary data.

Table 1: Clinical data from patients in the 1st trimester,

3rd trimester and preeclampsia groups

Comment 3: line 275 “Emerging study has suggested that GTA exerts an anti-tumor effect in breast cancer [19]. So we speculated that excessive GalNAc α 1,3 Gal and GTA in the placenta may contribute to the cause of preeclampsia”. Please clarify which is the connection with anti-tumor effect in breast cancer and the cause of PE.

Answer 3: Thank you for your valuable comment. The research has demonstrated that the invasion of trophoblasts into the uterus and the development of the placenta are similar to cancer cells invasion into the metastatic tissues or organs to a certain extent. At present, the cause of PE is mainly considered as shallow invasion of the trophoblast and insufficient remodeling of the uterine spiral artery. Study related to GTA expression and regulation in preeclampsia has not been reported. Here, we found that GTA was elevated in preeclampsia. However, GTA in association with tumor was less studied. Increased GTA inhibited invasion and metastasis ability of breast cancer cells, exerting an anti-tumor effect. Therefore, we speculated that GTA caused preeclampsia may partially impair trophoblast cells invasion and migration ability.

Comment 4: Line 333 "Placental tissue samples of the 3rd trimester and preeclampsia in the normal pregnancy (≥28 weeks) were collected from cesarean section pregnant women who were hospitalized for delivery" must be rephrased and clarified.

Answer 4: Thank you for the comments. We rephrased and clarified the placental tissue samples in revised manuscript. The content as follows:

Placental tissues were collected from 20 pregnant women who received perinatal care at Hospital of Dalian Medical University. The pregnancy outcome was determined in accordance with the definition of PE in the 25th edition of Williams Obstetrics, and the American College of Obstetricians and Gynecologists guidelines. PE was defined as a rise in systolic blood pressure to≥140mmHg or in diastolic pressure to ≥90mmHg on two separate occasions in a patient who was previously normotensive, as well as proteinuria of≥300 mg in a 24-hour collection, occurring after 20 weeks of pregnancy. Considering that the gestation period of PE is usually earlier than full term, we collected placentas earlier than 37 weeks as 3rd trimester, from healthy pregnancy with normal biochemical items and physical exam, excluding preeclampsia symptoms.

Reviewer 3 Report

Comments and Suggestions for Authors

The authors of this manuscript have tried to elucidate the underlying glycosylated related mechanism during the trophoblastic cell invasion in preeclampsia (PE). They found increased level of terminal GalNAc-a1, 3 Gal in blood group in placental trophoblast cells from preeclampsia patient as compared to healthy individuals with the help of soybean agglutinin lectin. This work demonstrates the importance of GTA mediated overexpression of GalNAc-a 1,3 Gal those results in dysregulated migration an invasion of extravillous trophoblast cells (EVTs). However, it is hard to make any conclusions as all the primary figures are missing in the paper.

I think the authors have somehow missed to include the figures while submitting the paper to the journal. I would recommend authors to re submit the paper with figures.

Also, include the catalog # for the reagents in the material and methods section.

Author Response

Thank you for reviewing our paper (Title: α1,3 N-acetylgalactosaminyl transferase (GTA) influences migration, invasion and proliferation of trophoblast cells in preeclampsia. Manuscript ID: ijms-2664419). We have carefully revised the manuscript following your comments and suggestions, which are very helpful in improving this manuscript. The point-to-point answers are as follows:

The authors of this manuscript have tried to elucidate the underlying glycosylated related mechanism during the trophoblastic cell invasion in preeclampsia (PE). They found increased level of terminal GalNAc-a1, 3 Gal in blood group in placental trophoblast cells from preeclampsia patient as compared to healthy individuals with the help of soybean agglutinin lectin. This work demonstrates the importance of GTA mediated overexpression of GalNAc-a 1,3 Gal those results in dysregulated migration an invasion of extravillous trophoblast cells (EVTs). However, it is hard to make any conclusions as all the primary figures are missing in the paper.

Comment 1: I think the authors have somehow missed to include the figures while submitting the paper to the journal. I would recommend authors to re submit the paper with figures.

Answer 1: Thank you for your reminding. Given that the reviewer did not see the complete figures, we have uploaded all the data again, and made sure the uploaded data were complete.

Comment 2: Also, include the catalog # for the reagents in the material and methods section.

Answer 2: Thank you. Catalog # for the reagents have been added in the materials and methods section in revised manuscript.
